# Isolation and Characterization of Flavonoids from Fermented Dandelion (*Taraxacum mongolicum* Hand.-Mazz.), and Assessment of Its Antioxidant Actions In Vitro and In Vivo

**Na Yin** [1,2,†] **, Yuan Wang** [1,2,†] **, Xuerong Ren** [1,2] **, Yang Zhao** [1,2] **, Na Liu** [1,2] **, Xiaoping An** [1,2,*] **and Jingwei Qi** [1,2,*]

1. College of Animal Science, Inner Mongolia Agricultural University, Hohhot 010018, China; yinna0420@163.com (N.Y.); wangyuan.926@163.com (Y.W.); r2157162@163.com (X.R.); zyxqx03@163.com (Y.Z.); liuna_dky@163.com (N.L.)
2. Inner Mongolia Herbivorous Livestock Feed Engineering Technology Research Center, Inner Mongolia Agricultural University, Hohhot 010018, China
* Correspondence: anxiaoping@imau.edu.cn (X.A.); qijingwei@imau.edu.cn (J.Q.)
† These authors contributed equally to this work.

**Abstract:** Flavonoids are famous for their diverse sources, strong biological activity, and low toxicity and could be used as a natural antioxidant in animal husbandry. In this study, the purification process and antioxidant activity of flavonoids from fermented dandelion were investigated. The adsorption and desorption characterizations of AB-8 macroporous resin for flavonoids from fermented dandelion (FD) were determined and purification parameters were optimized. Qualitative analysis using UPLC-MS/MS analysis was explored to identify the components of the purified flavonoids of FD (PFDF). The antioxidant activity of PFDF in vitro and in vivo was analyzed. The optimum purification parameters were as follows: a sample concentration of 2 mg/mL, 120 mL of the sample volume, a pH of 2.0, and eluted with 90 mL of 70% ethanol (pH 5). After purification, the concentration of the flavonoids in PFDF was 356.08 mg/mL. By comparison with reference standards or the literature data, 135 kinds of flavonoids in PFDF were identified. Furthermore, PFDF had a strong reducing power and scavenging ability against 8-hydroxy radical and DPPH radical. PFDF can effectively reduce the oxidative stress of zebrafish embryos and IPCE-J2 cells by modulating antioxidant enzyme activities. In summary, the purified flavonoids from fermented dandelion have good antioxidant activity and display superior potential as a natural antioxidant in animal husbandry.

**Keywords:** fermented dandelion; flavonoids; purification; characterization; antioxidant activity

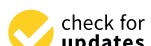



## 1. Introduction

Reactive oxygen species (ROS) are crucial to the maintenance of cellular homeostasis, which is achieved by a system of antioxidant defenses. The antioxidant defense system maintains a balance between oxidative stress and antioxidant protection [1,2]. Excess ROS can attack and damage various biomolecules such as DNA, proteins, carbohydrate, and membrane lipids leading to cell death and tissue damage. Thus, it is critically important to reduce ROS generation.

The secondary metabolite compounds derived from plants could act as natural antioxidants in preventing oxidative damage. Among them, flavonoids are especially effective in capturing superoxide anion radicals, hydroxyl radicals, singlet oxygen, and lipid radicals [3]. Furthermore, flavonoids have been demonstrated to inhibit the activity of xanthine oxidase, which catalyzes the generation of superoxide radical anion [4]. Dandelion (*Taraxacum mongolicum* Hand.-Mazz., Asteraceae family), as a widely distributed Chinese herb, exhibits antioxidative, antiinflammatory, antimicrobial, and immunostimulatory activities. Hagymasi et al. (2000) [5] found that the extracts of dandelion leaves and roots can inhibit the formation of ROS and scavenge free radicals. Another study also found that the

ethyl acetate component of dandelion flower extracts scavenged ROS and protected DNA from ROS-induced damage [6]. More than 30 phenolic compounds have been identified and isolated from dandelion. It has been demonstrated that the antioxidant property of dandelion is attributed to the high abundance of phenolic components such as flavonoids, coumaric acid, and ascorbic acid [7].

Microbial fermentation technology has been widely used in the separation of bioactive components from plants. During fermentation, the components of raw materials are broken down and new useful molecules are formed; therefore, the content of some bioactive substances is increased [8] In our preliminary research, we found that the content of flavonoids in fermented dandelion (FD) crude extract was significantly higher than that of raw dandelion crude extract. In addition, FD crude extract displayed superior antioxidant activity to that of raw dandelion crude extract in vitro [9]. To go one step further, the flavonoids from FD should be identified and isolated to optimize and promote the development of new food additives.

Therefore, in the present study, we purified and characterized the flavonoids from FD, and analyzed the antioxidant activity of the purified flavonoids of FD (PFDF) in IPEC-J2 cells and zebrafish embryos, aiming to demonstrate the health benefits of the flavonoids in FD and expand their use in functional food.

## 2. Materials and Methods

### 2.1. Materials and Reagents

Dandelion was purchased from local market. The *Saccharomyces cerevisiae* (CGMCC 2.1190) and *Lactobacillus plantarum* (CGMCC 1.12934) were obtained from Inner Mongolia Herbivorous Livestock Feed Engineering and Technology Research Center. Yeast extract, ethanol, nutrient broth medium, Wort medium, sodium nitrite ($NaNO_2$), aluminum nitrate ($Al(NO_3)_3$), sodium hydroxide (NaOH), rutin, DPPH (2,2-diphenyl-1-picrylhydrazyl), ferrous sulfate ($FeSO_4$), disodium hydrogen phosphate ($Na_2HPO_4$), sodium dihydrogen phosphate ($NaH_2PO_4$), hydrogen peroxide ($H_2O_2$), salicylic acid, potassium ferricyanide, trichloroacetic acid, ferric chloride ($FeCl_3$), 2,2-diphenyl-1-picrylhydrazyl (DPPH), 2,7-dichlorofluorescein diacetate (DCF-DA), diphenyl-1-pyrenylphosphine (DPPP), acridine orange (AO), and dimethyl sulfoxide (DMSO) were purchased from Sigma (St. Louis, MO, USA). Butylated hydroxyanisole (BHA) was purchased from Sinopharm Chemical Reagent Co. (Beijing, China). DMEM/F12 medium, fetal bovine serum (FBS), insulin transferrin selenium (ITS), penicillin, and streptomycin were purchased from GIBCO (Grand Island, NE, USA). Cell Counting Kit-8 (CCK-8) was purchased from Solarbio (Beijing, China). All other chemicals and reagents were analytical grade.

### 2.2. Preparation of FD Extract

The fermentation and extraction of dandelion were conducted following the method used by Liu et al. (2020) [9]. Briefly, *S. cerevisiae* and *L. Plantarum* were stirred at a ratio of 7:3, with a final concentration of $1 \times 10^8$ CFU/mL, to prepare inoculum. The dandelion was inoculated with 10% (*v/v*) inoculum and 0.5% yeast extract. The ratio of material to liquid was set at 1:1.2. The fermentation was conducted at 35 °C for 50 h. After fermentation, the substrate was dried by hot air and ground with a mill. The dried FD was extracted at 80 °C with distilled water at a ratio of 1:20 for 30 min. The supernatant was lyophilized, then extracted at 70 °C with 40% ethanol at a ratio of 1:35 for 30 min. The FD extract was dried and stored for further purification.

### 2.3. Purification of Flavonoids from FD Extract by Macroporous Resin

2.3.1. Pretreatment of Macroporous Resins

The AB-8 macroporous resin was steeped in 95% ethanol for 12 h before being rinsed with deionized water until no ethanol remained. The microporous resin was dried in a 60 °C oven for 4 h for subsequent use.

2.3.2. Static Adsorption and Desorption Tests

The initial sample solution concentrations (0.5, 1, 1.5, 2, 2.5, and 3 mg/mL) and pH values (2.0, 3.0, 4.0, 5.0, 6.0, 7.0, and 8.0) as well as the eluent ethanol concentrations (20%, 30%, 40%, 50%, 60%, 70%, and 80%) and pH values (1, 2, 3, 4, 5, 6, 7, and 8) were screened by static tests. FD solution (30 mL) was adsorbed by 2 g of macroporous resin, and the total flavonoid content was determined after 12 h of oscillation. After that, these macroporous resins were desorbed by 30 mL of ethanol solution, and the total flavonoid content was determined again after 12 h of oscillation.

2.3.3. Dynamic Adsorption and Desorption Tests

Dynamic tests were conducted in a glass column containing 20 g of macroporous resin, with 200 mL of sample solution (FD, 2.0 mg/mL, pH 3.0) flowing through at a rate of 1 mL/min. The total flavonoid contents were determined every 10 min. The adsorbed macroporous resin was rinsed with distilled water until the solution became colorless, then 200 mL of ethanol solution (70%, pH 5) was allowed to flow through the glass column at a flow rate of 1 mL/min. The total flavonoid content was determined again every 10 min.

The calculation formulas were as follows

$$Q = \frac{(C_0 - C_1)V_1}{W}, \tag{1}$$

$$A\ (\%) = \frac{(C_0 - C_1)}{C_0} \times 100,\ \text{and} \tag{2}$$

$$D\ (\%) \frac{C_2 V_2}{(C_0 - C_1)V_1} \times 100 \tag{3}$$

where Q was the adsorption capacity (mg/g resin); A was the adsorption ratio (%); $C_0$ reflected the initial concentrations of total flavonoid in the adsorption solution (mg/mL); $C_1$ referred to the equilibrium concentrations of flavonoid solutions (mg/mL); $V_1$ was the volume of the initial sample solution (mL); W was the weight of the dry resin (g); D was the desorption ratio (%); $C_2$ was the concentration of total flavonoid in the desorption solution (mg/mL); and $V_2$ was the volume of the desorption solution (mL).

*2.4. Determination of Total Flavonoid Content*

The determination of total flavonoids' content was conducted with a previously reported method by Zhang et al. (2011) [10] with some modifications. The sample solution (50 mL) was mixed with 0.4 mL of $NaNO_2$ for 6 min, 0.4 mL of $Al\ (NO_3)_3$ was added to the mixture for 6 min, and then 4 mL of NaOH was added for 15 min. The total volume was adjusted to 10.0 mL with distilled water, and the absorbance was measured at 510 nm. The total flavonoid content was determined using a standard curve of rutin. The calibration curve was y = 5.7929x + 0.0449, where y was the absorbance value of the sample and x was the sample concentration ($r^2$ = 0.9997).

*2.5. Determination PFDF Composition*

Non-targeted metabolomics analysis using ultra-performance liquid chromatography–tandem mass spectrometry (UPLC-MS/MS) is a rapid and highly sensitive method for detecting plant metabolites and is based on information present in scientific databases [11]. The chemical identification of metabolic characteristics discovered in data sets recorded on LC-MS systems often begins with an accurate measurement of the m/z, and observations of the m/z can be utilized to match a metabolic feature to a single or limited number of molecular formulae. Following the identification of single or multiple molecular formulas, they were matched to known metabolites by searching an array of online or laboratory-specific resources (including MassBank (http://www.massbank.jp/ (accessed on 17 December 2019)), KNAPSAcK (http://kanaya.naist.jp/KNApSAc/ (accessed on 17 December 2019)), HMDB (http://www.hmdb.ca/ (accessed on 17 December 2019)) [12],

MoTo DB (http://www.ab.wur.nl/moto/ (accessed on 17 December 2019)), METLIN (http://metlin.scripps.edu/index.php/ (accessed on 17 December 2019)) [13], and MWDB (metware database); and the content of relevant components in the samples was calculated based on the retention times of metabolites and the intensity of ion flow from the ion detection of PFDF. The analytical conditions were as follows: HPLC column, Waters ACQUITY UPLC HSS T3 C18 (1.8 μm, 2.1 mm × 100 mm); solvent system, water (0.04% acetic acid)–acetonitrile (0.04% acetic acid); gradient program, 100:0 $v/v$ at 0 min, 5:95 $v/v$ at 11.0 min, 5:95 $v/v$ at 12.0 min, 95:5 $v/v$ at 12.1 min, and 95:5 $v/v$ at 15.0 min; flow rate, 0.40 mL/min; temperature, 40 °C; and injection volume, 5 μL. The mass spectrum peaks of all substances were integrated by peak area after receiving the mass spectrum analysis data of PFDF metabolites, and the mass spectrum peaks of the same metabolite in various samples were integrated and corrected [14].

### 2.6. In Vitro Antioxidant Activity of PFDF

2.6.1. Reducing Power

Analysis of reducing power was performed by the method of Yildirim et al. (2001) [15] with slight changes. Briefly, 0.75 mL of phosphate buffer (200 mM, pH = 6.6) and 0.75 mL of potassium ferricyanide (1%) were mixed with 0.75 mL of sample solution for 20 min at 50 °C, then, 0.75 mL of trichloroacetic acid was added to terminate the reaction. Then 1.5 mL of distilled water and 400 μL of $FeCl_3$ (0.1%) were added to 1.5 mL of mixture for 10 min at room temperature; the absorbance was measured at 700 nm using a BioTek microplate reader (BioTek Instruments Inc., Winooski, VT, USA).

2.6.2. DPPH Radical Scavenging Activity

The determination of DPPH radical scavenging activity was slightly modified from a previously reported method [16]. Two milliliters of sample solution were added to 2 mL of DPPH solution for 30 min, and the OD was measured at 517 nm.

The calculation formula was as follows

$$DPPH - radical\ scavenging\ activity\ (\%) = \left(1 - \frac{A_0 - A_1}{A_2}\right) \times 100, \tag{4}$$

where $A_0$ was the absorbance of the sample mixed with DPPH; $A_1$ was the absorbance of the sample mixed with 95% ethanol solution; $A_2$ was the absorbance of DPPH solution with 95% ethanol solution.

2.6.3. Hydroxyl Radical Scavenging Activity

The hydroxyl radical scavenging activity was determined using the method of Rajauria et al. (2012) [17] with some modifications. For 10 min at room temperature, 0.5 mL of test sample solution was mixed with 0.5 mL of $FeSO_4$ (9 mmol/L) and 0.5 mL of $H_2O_2$ (8.8 mmol/L), followed by 0.5 mL of salicylic acid solution (9.0 mmol/L) for 30 min at room temperature. Absorbance was determined at 510 nm.

The calculation formula was as follows

$$Hydroxyl - radical\ scavenging\ activity\ (\%) = \left(1 - \frac{A_1 - A_2}{A_0}\right) \times 100, \tag{5}$$

where $A_0$ was the absorbance of sample solution replaced by distilled water; $A_1$ was the absorbance of sample solution; $A_2$ was the absorbance of $FeSO_4$ solution replaced by distilled water.

### 2.7. Zebrafish Embryo Antioxidant Activity of PFDF

2.7.1. Maintenance of Parental Zebrafish

Adult zebrafish were bought from the China Zebrafish Resource Center in Wuhan, China, and kept in an acrylic tank at a temperature of 28.5 °C with a 14/10-h light/dark

cycle. Commercial fish feed enriched with live *Artemia salina* was fed twice a day to the zebrafish. Spawning was stimulated by turning on the lights in the morning, and the embryo collecting took only 1 h.

### 2.7.2. Exposure of Zebrafish Embryos to PFDF

At 7 hpf (hours after fertilization), the embryos were transferred to 24-well plates and subjected to PFDF at 0 (control), 0.3125, 0.625, 1.25, and 2.5 μg/mL. The concentration range of PFDF in this experiment was obtained from preliminary experiment. At 24 hpf, the embryos were cleaned three times in fresh embryo media [18]. Up to 72 hpf, the zebrafish larvae hatched from the embryos were used for further intracellular ROS production, cell death, and lipid peroxidation measurements.

### 2.7.3. Intracellular ROS Production, Cell Death and Lipid Peroxidation Measurements and Image Analysis

The fluorescent probe dyes DCFH-DA, AO, and DPPP were used to examine intracellular ROS production, cell death, and lipid peroxidation in zebrafish larvae. The zebrafish larvae were put into 24-well plates and treated with DCF-DA (20 g/mL), AO (7 g/mL), and DPPP (25 g/mL) solutions. The plates were incubated in the dark at 28.5 °C for 1 h, 0.5 h, and 1 h, respectively. The zebrafish larvae were monitored with a CoolSNAP-Procolor digital camera (Olympus, Japan) according to Kim et al. (2016) [19]. The fluorescence intensity of the individual zebrafish larvae was quantified using the ImageJ program (NIH, USA).

### 2.7.4. Determination of CAT, SOD, GSH-Px and MDA in Zebrafish Embryos

PFDF exposure was performed as described in Section 2.7.1. At 16 hpf, juvenile zebrafish embryos in 24-well plates were washed with fresh embryo media, lysed in PBS, then transferred to 1.5-mL pre-weighed centrifuge tubes, quickly weighed, and frozen ($-80$ °C) for catalase (CAT), superoxide dismutase (SOD), glutathione peroxidase (GSH-Px), and malondialdehyde (MDA) determination with the assay kits. The 240 embryos (6 pools of 40 eggs per treatment) were collected.

### *2.8. Cellular Antioxidant Activity of PFDF*

### 2.8.1. Cell Culture and Treatments

The cell line IPEC-J2 was acquired from College of Animal Science and Technology, China Agricultural University. The IPEC-J2 cells were grown at 37 °C in DMEM/F12 media with 10% (*v/v*) FBS, 1% ITS, and 1% penicillin–streptomycin in an incubator with 5% $CO_2$ (*v/v*). IPEC-J2 cells were seeded in 96-well plates at a density of $1 \times 10^5$ cells/well and cultured to 80% confluence. Then, cells were incubated with PFDF at the final concentrations of 0 (control), 0.625, 1.25, 2.5, 5, and 10 μg/mL for 24 h.

### 2.8.2. Cell Proliferative Activity Measurements

The proliferative activity of IPEC-J2 cells was measured using the CCK8 kit according to the manufacturer's instructions. After incubation as the method of Section 2.8.1, 10 μL of CCK8 solution was supplemented and incubated at 37 °C for 2 h. Finally, absorbance was monitored at 450 nm using a microplate reader.

The calculation formula was as follows

$$\text{Proliferative activity } (\%) = \left(1 - \frac{A_1 - A_0}{A_2 - A_0}\right) \times 100, \tag{6}$$

where $A_1$ was the absorbance of PFDF-treated cells; $A_2$ was the absorbance of untreated cells; $A_0$ was the absorbance of control.

### 2.8.3. Determination of CAT, SOD, GSH-Px, GSH and MDA in IPEC-J2 Cells

After incubation according to Section 2.8.1, cells were scraped, sonicated in cell lysis buffer, centrifuged at $10,000 \times g$ for 10 min. The supernatant was collected for further

analysis. Commercial assay kits were used to determine MDA and glutathione (GSH) levels, as well as antioxidative enzyme activity (GSH-Px, CAT, and SOD) in IPEC-J2 cells. The BCA protein assay kit was used to determine the protein concentration of the supernatant.

*2.9. Statistical Analysis*

All of the tests were carried out in triplicate. The data were submitted to one-way analysis of variance (ANOVA) using SAS and were presented as means standard error (SE). Tukey's test was used to compare the means of treatments that exhibited significant differences. The significance threshold was chosen at $p < 0.05$.

## 3. Results and Discussion

*3.1. Purification of Flavonoids FD Extract by AB-8 Macroporous Resin*

3.1.1. Static Adsorption and Desorption

Through hydrogen bonding interactions and van der Waals forces, macroporous resin can be utilized to selectively adsorb components from aqueous solutions [20]. Here, we optimized the FD flavonoids' purification conditions with AB-8 macroporous resin. As shown in Figure 1A, the adsorption rate of the flavonoids increased with the initial sample concentration and reached its peak value at 2 mg/mL; then, it decreased gradually. This finding may be attributed to the quality of the miscellaneous competitive adsorption with flavonoids rising as the sample solution concentration increased. Therefore, the suitable initial sample concentration was 2 mg/mL. Flavonoids are weak acids and have a phenolic hydroxyl structure; therefore, the pH of the sample solution directly affects the adsorption properties of macroporous resins. As can be seen from Figure 1B, the adsorption rate of the flavonoids at a pH of 2.0 (94.33%) was significantly higher than the other initial pH value; when the pH value exceeded 2.0, the absorption ratio of the flavonoids decreased dramatically. At higher pH values, the hydrogen bonding interactions are reduced because the phenolic hydroxyl groups in flavonoids dissociate to form $H^+$ and their corresponding anions, resulting in the decrease in the adsorption rate. Thus, the optimal sample solution pH value was adjusted to 2.0 for the later tests.

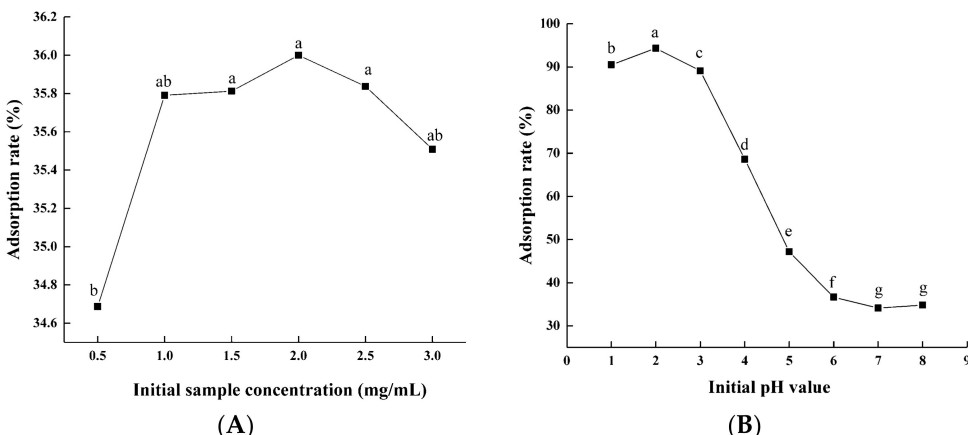

**Figure 1.** Effects of initial sample solution concentration (**A**) and pH value (**B**) on the adsorption rate of macroporous resin. Experiments were performed in triplicate, and data are expressed as the mean $\pm$ SE. Different lowercase superscript letters denote statistically significant differences ($p < 0.05$).

The desorption of the flavonoids on the macroporous resins was due to the competition between the intermolecular forces of adsorption on the macroporous resins and dissolution in the solvent. When the intermolecular force decreased, the flavonoids were desorbed from the macroporous resins into the solvent. As shown in Figure 2A, the desorption rate of the flavonoids increased with of the rise in ethanol concentrations. The desorption ratio of the flavonoids was the highest when the concentration of ethanol was 70% and then leveled

off. Figure 2B shows that with increasing ethanol pH value, the desorption ratio of the flavonoids increases accordingly. The maximum desorption ratio was found when using ethanol at a pH of 5.0. It was obvious that 70% ethanol (pH 5.0) had a higher capacity to decrease the affinity between the flavonoids and macroporous resins, thus facilitating more efficient desorption. Therefore, we designated 70% as the optimal ethanol concentration and 5.0 as the optimal pH.

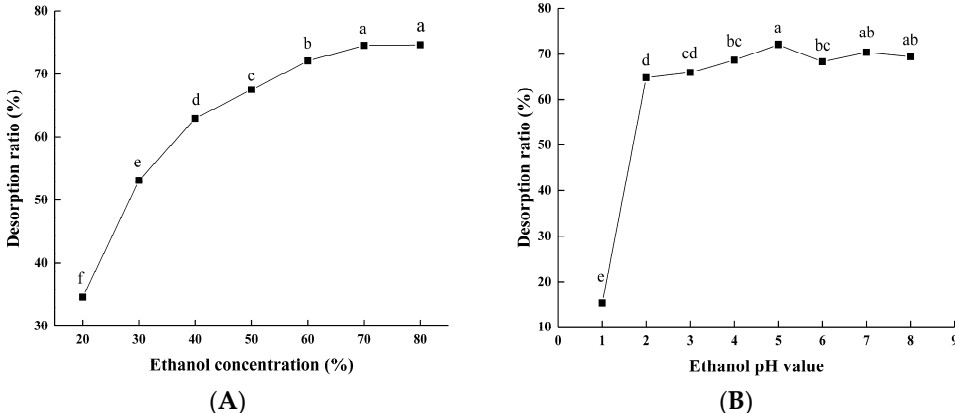

**Figure 2.** Effects of ethanol concentration (**A**) and pH value (**B**) on desorption rate of macroporous resin. Experiments were performed in triplicate, and data are expressed as the mean $\pm$ SE. Different lowercase superscript letters denote statistically significant differences ($p < 0.05$).

### 3.1.2. Dynamic Adsorption and Desorption

It was found that when the adsorption reaches the break point, the adsorption effect decreases, and the solutes leak from the resin. As can be seen from Figure 3A, when the volume of the initial sample solution was lower than 120 mL, almost all flavonoids were absorbed by the AB-8 macroporous resin. When the volume exceeded 120 mL, the concentration of the total flavonoids in the leakage solution increased rapidly until it stabilized at 200 mL. Therefore, to prevent sample loss, the volume of the initial sample solution was set at 120 mL. With the increase in eluent volume, the concentration of the total flavonoids in the eluent increased first and then decreased (Figure 3B). When the eluent volume reached 90 mL, the flavonoids were almost completely eluted. Therefore, 90 mL was chosen as the optimal eluent volume.

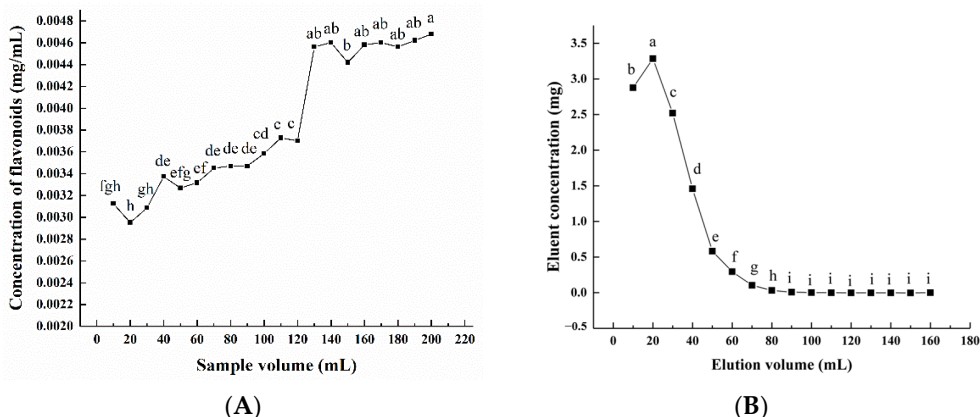

**Figure 3.** Effect of sample volume (**A**) and elution volume (**B**) on the desorption rate of macroporous resin. Experiments were performed in triplicate, and data are expressed as the mean $\pm$ SE. Different lowercase superscript letters denote statistically significant differences ($p < 0.05$).

Thus, these results revealed that the optimization parameters of the purification process of flavonoids on AB-8 macroporous resin were a sample concentration of 2 mg/mL,

pH of 2.0, and sample volume of 120 mL; an elution solvent ethanol–water (70%), pH of 5, and eluent volume of 90 mL. After purification, the concentration of the total flavonoids in PFDF was 356.08 mg/mL.

*3.2. Composition of PFDF*

　　　Flavonoids generally refer to a class of compounds formed by the connection of two benzene rings and with 2-phenylchromone as the basic nucleus [21]. Flavonoid glycosides with different structures are classified according to the type, number, and location of sugars. A small portion of the flavonoids isolated from natural products are free aglycones, i.e., flavonoid aglycones; however, most of these are glycosidic forms formed by flavonoid aglycones and sugars. Using UPLC-MS/MS, we analyzed the composition of PFDF, and 135 flavonoid compounds were isolated and identified. There are 69 main flavonoid glycosides, including flavonoid, flavonols, flavonoid carbonoside, dihydroflavone, dihydroflavonol, and isoflavones (Table S1), and 16 main flavonoid aglycones (Table S2). Many flavonoid glycosides were detected in PFDF, which may be due to the water extraction of FD as most flavonoid glycosides are more soluble in water than flavonoid aglycones.

　　　The 10 main flavonoid glycosides (Table 1, Figure 4) and flavonoid aglycones (Table 2, Figure 5) were shown. It was found that luteolin is the substance with the highest relative content detected in PEDF. Nicotiflorin (kaempferol 3-*O*-rutinoside), kaempferol-3-*O*-neohesperidin, Kaempferol 3-*O*-robinobioside (Biorobin), and kaempferol-3-*O*-glucoside-7-*O*-rhamnoside are glycoside derivatives of kaempferol, and their relative contents are relatively high in PFDF. These flavonoid glycosides contain three hydroxyl groups. It is found that the more $H^+$ that can be ionized, the more free radicals can be bound and stronger antioxidant capacity exhibited. From the primary flavonoid aglycones in PFDF, the compounds with a relatively higher content included luteolin, hispidulin, diosmetin, taxifolin, and tricin. Previous study has shown that flavonoid aglycones are more easily absorbed and utilized than flavonoid glycosides and exhibit strong antioxidant activity [22]. Therefore, we infer that PFDF could exhibit strong antioxidant activity due to their higher flavonoid glycoside and aglycones content.

**Table 1.** The relative content of major flavonoid glycosides.

| | Molecular Weight (Da) | Formula | Compounds | Class II | Relative Content |
|---|---|---|---|---|---|
| $A_1$ | 449.1 | $C_{21}H_{20}O_{11}$ | Luteolin 7-*O*-glucoside (Cynaroside) | Flavonoid | 23,420,000 |
| $A_2$ | 593.16 | $C_{27}H_{30}O_{15}$ | Kaempferol 3-*O*-rutinoside (Nicotiflorin) | Flavonols | 21,362,000 |
| $A_3$ | 449.11 | $C_{21}H_{20}O_{11}$ | Quercetin-3-*O*-α-L-rhamnopyranoside | Flavonols | 20,955,000 |
| $A_4$ | 595.16 | $C_{27}H_{30}O_{15}$ | Kaempferol-3-*O*-neohesperidoside | Flavonols | 17,111,000 |
| $A_5$ | 595.16 | $C_{27}H_{30}O_{15}$ | Luteolin-7-*O*-rutinoside | Flavonoid | 16,898,000 |
| $A_6$ | 593.16 | $C_{27}H_{30}O_{15}$ | Kaempferol 3-*O*-robinobioside (Biorobin) | Flavonols | 16,529,000 |
| $A_7$ | 595.16 | $C_{27}H_{30}O_{15}$ | Kaempferol-3-*O*-glucoside-7-*O*-rhamnoside | Flavonols | 13,746,000 |
| $A_8$ | 595.17 | $C_{27}H_{30}O_{15}$ | Tetrahydroxyflavone-C-rhamnosyl-glucoside | Flavonoid carbonoside | 12,264,000 |
| $A_9$ | 595.16 | $C_{27}H_{30}O_{15}$ | Lonicerin | Flavonoid | 12,034,000 |
| $A_{10}$ | 435.08 | $C_{20}H_{18}O_{11}$ | Avicularin | Flavonols | 11,444,000 |

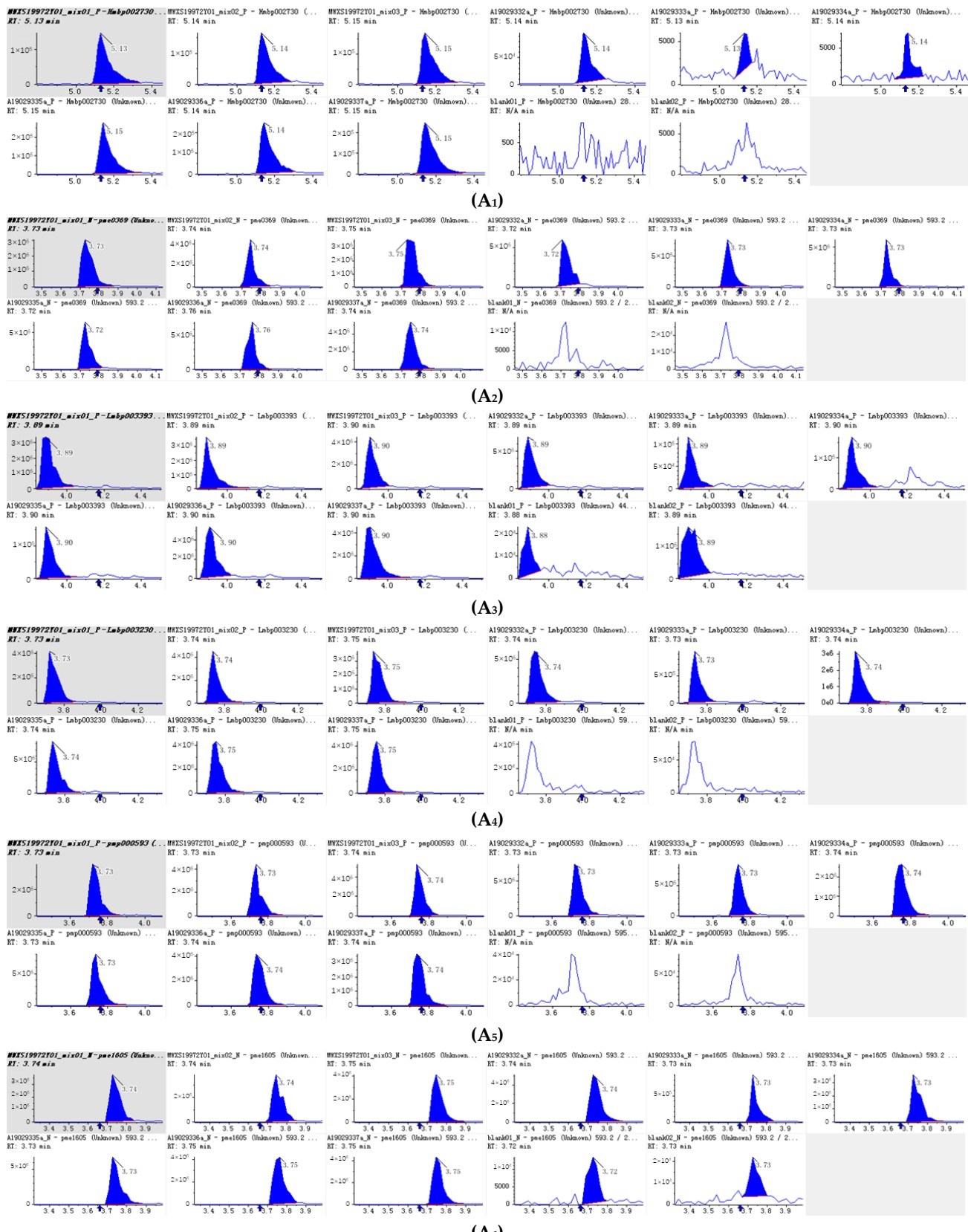

**Figure 4.** *Cont.*

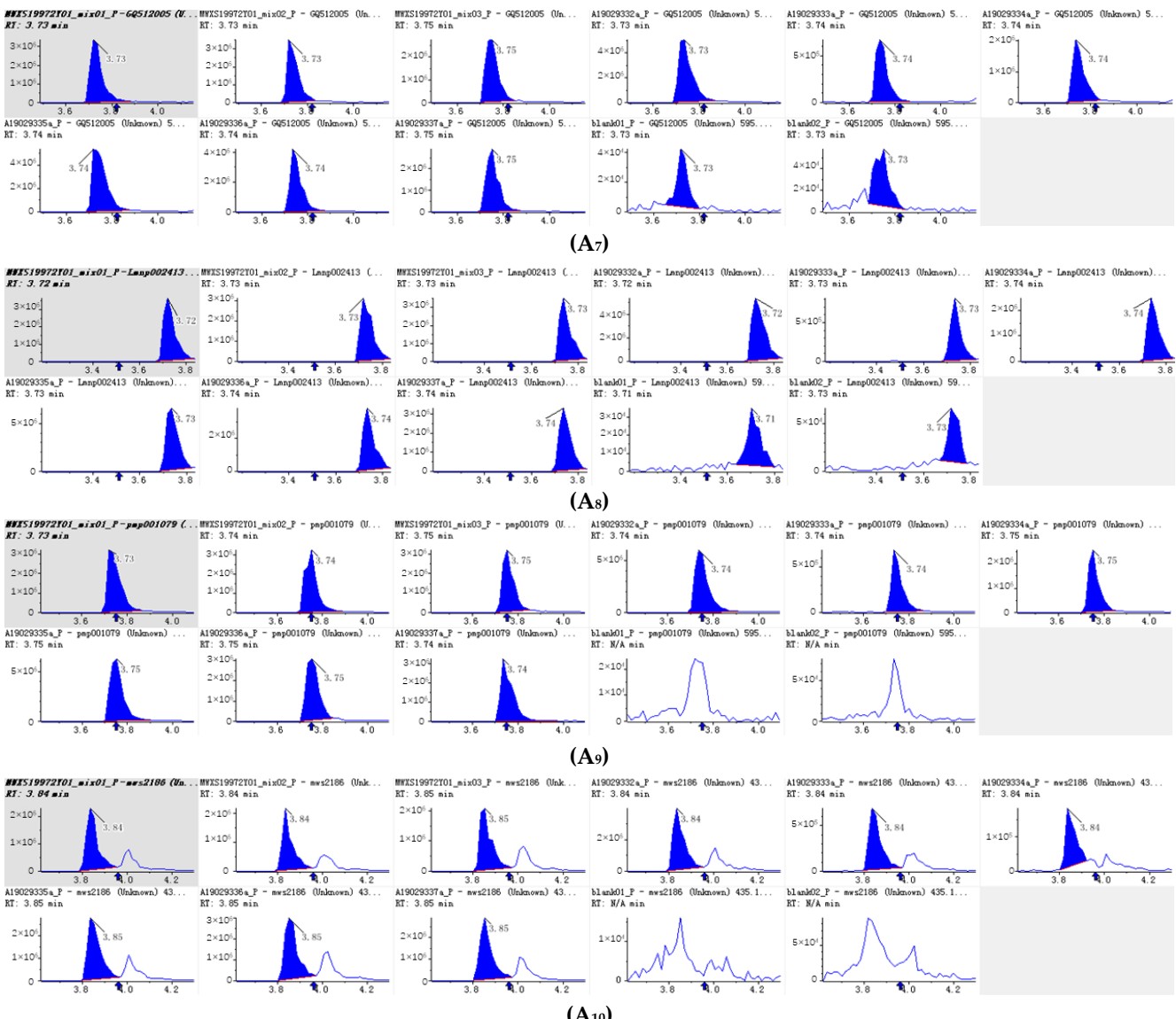

**Figure 4.** Integral diagram of Luteolin 7-O-glucoside (Cynaroside) (**A₁**), Kaempferol 3-O-rutinoside (Nicotiflorin (**A₂**), Quercetin-3-*O*-α-L-rhamnopyranoside (**A₃**), Kaempferol-3-*O*-neohesperidoside (**A₄**), Luteolin-7-*O*-rutinoside (**A₅**), Kaempferol 3-*O*-robinobioside (Biorobin (**A₆**), Kaempferol-3-*O*-glucoside-7-*O*-rhamnoside (**A₇**), Tetrahydroxyflavone-C-rhamnosyl-glucoside (**A₈**), Lonicerin (**A₉**) and Avicularin (**A₁₀**) quantitative analysis in PFDF.quantitative analysis in PFDF. The abscissa is the retention time of metabolites (min), the ordinate is the ion current intensity of metabolites (cps), and the peak area represents the relative content of substances in the sample. The arrow in the figure indicates that the position of retention time corresponds to the RT value in the figure.

**Table 2.** The relative content of major flavonoid aglycones.

| | Molecular Weight (Da) | Formula | Compounds | Class II | Relative Content |
|---|---|---|---|---|---|
| $B_1$ | 286.04 | $C_{15}H_{10}O_6$ | Luteolin | Flavonoid | 4,983,700 |
| $B_2$ | 300.05 | $C_{16}H_{12}O_6$ | Hispidulin | Flavonoid | 3,514,700 |
| $B_3$ | 300.05 | $C_{16}H_{12}O_6$ | Diosmetin | Flavonoid | 3,472,700 |
| $B_4$ | 304.05 | $C_{15}H_{12}O_7$ | Taxifolin | Dihydroflavonol | 3,057,300 |
| $B_5$ | 330.07 | $C_{17}H_{14}O_7$ | Tricin | Flavonoid | 1,137,200 |
| $B_6$ | 286.05 | $C_{15}H_{10}O_6$ | Isoscutellarein | Flavonoid | 1,030,300 |
| $B_7$ | 330.07 | $C_{17}H_{14}O_7$ | Jaceosidin | Flavonoid | 522,290 |
| $B_8$ | 372.11 | $C_{20}H_{20}O_7$ | Tangeretin | Flavonols | 500,730 |
| $B_9$ | 274.07 | $C_{15}H_{14}O_5$ | Phloretin | Chalcones | 498,600 |
| $B_{10}$ | 448.08 | $C_{21}H_{20}O_{11}$ | Isoorientin | Flavonoid carbonoside | 436,490 |

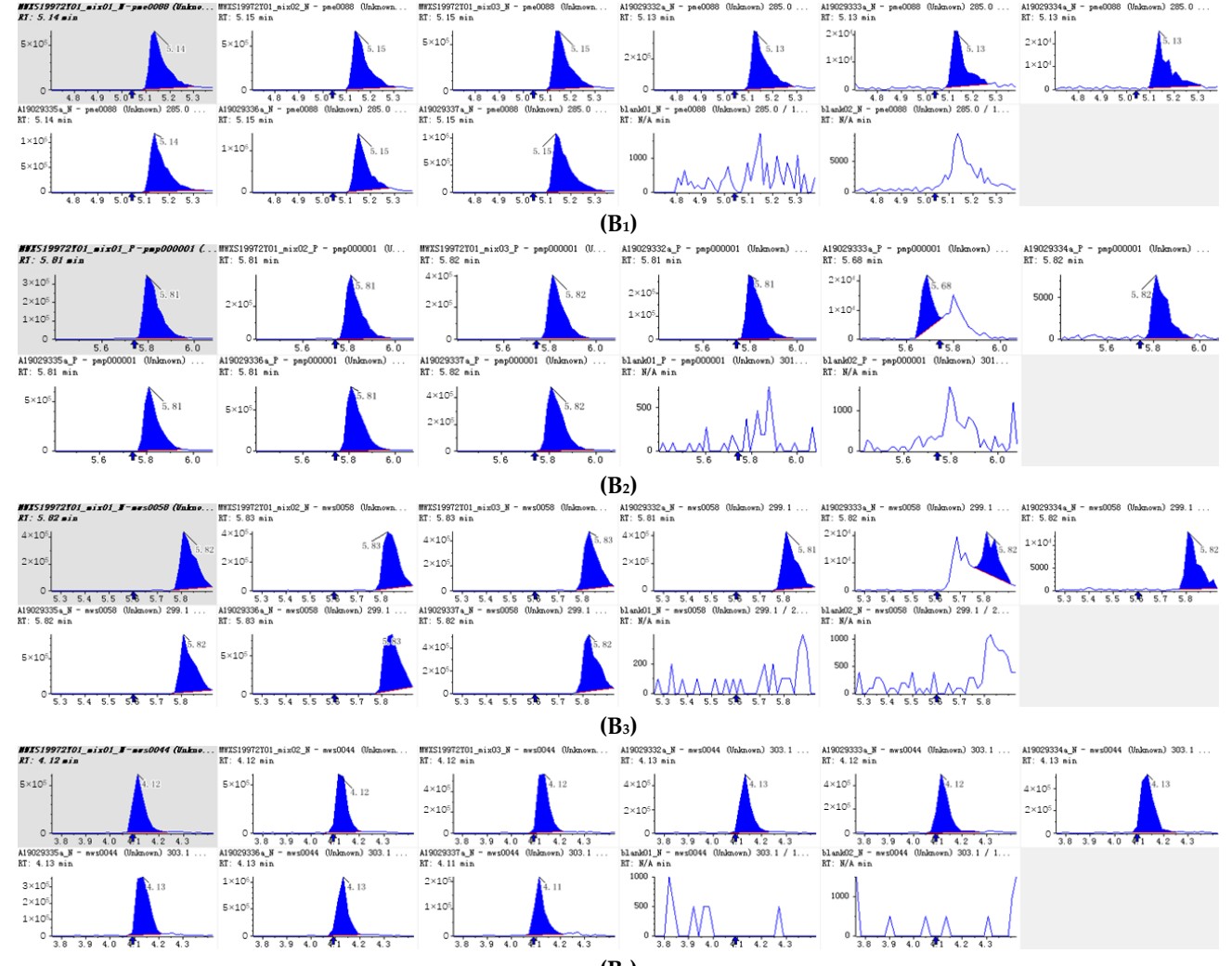

**Figure 5.** *Cont.*

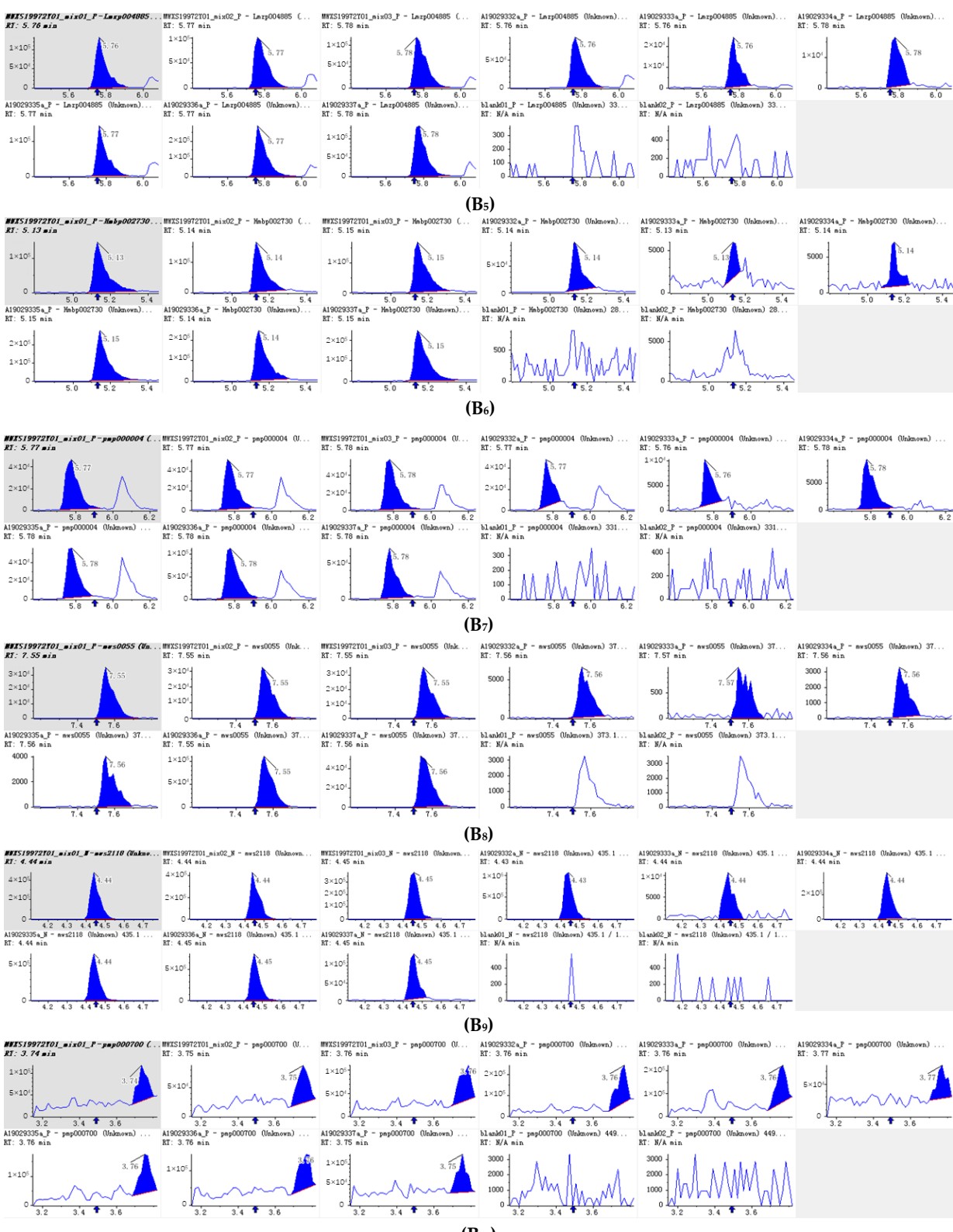

**Figure 5.** Integral diagram of Luteolin (**B₁**), Hispidulin (**B₂**), Diosmetin (**B₃**), Taxifolin (**B₄**), Tricin (**B₅**), Isoscutellarein (**B₆**), Jaceosidin (**B₇**), Tangeretin (**B₈**), Phloretin (**B₉**) and Isoorientin (**B₁₀**) quantitative analysis in PFDF.The abscissa is the retention time of metabolites (min), the ordinate is the ion current intensity of metabolites (cps), and the peak area represents the relative content of substances in the sample. The arrow in the figure indicates that the position of retention time corresponds to the RT value in the figure.

### 3.3. In Vitro Antioxidant Activity of PFDF

DPPH is a stable free radical with a maximum absorbance of 517 nm and is commonly employed as a reagent to test the free-radical scavenging ability of antioxidants [23]. The absorbance is lowered when DPPH radicals make contact with a proton donor substrate such as an antioxidant and are scavenged [24,25]. We observed that the DPPH free-radical scavenging capacity of PFDF increased in a dose-dependent manner (Figure 6A). When the concentration was between 0.015 and 0.045 mg/mL, the DPPH scavenging capacity of PFDF was slightly lower than that of vitamin C (VC), while at 0.06–0.075 mg/mL, the PFDF exhibited comparable DPPH free-radical scavenging activity to that of VC. When the concentration reached 0.075 mg/mL, the scavenging rates of PFDF and VC were 94.57% and 96.27%, respectively. Flavonoids are weak acids and have a phenolic hydroxyl structure. It is suggested that the phenolic hydroxyl groups in flavonoids could donate H+ and exhibit free-radical scavenging ability. Sheikh et al. (2015) [26] found similar results, showing that dandelion leaves had strong DPPH free-radical scavenging capacity.

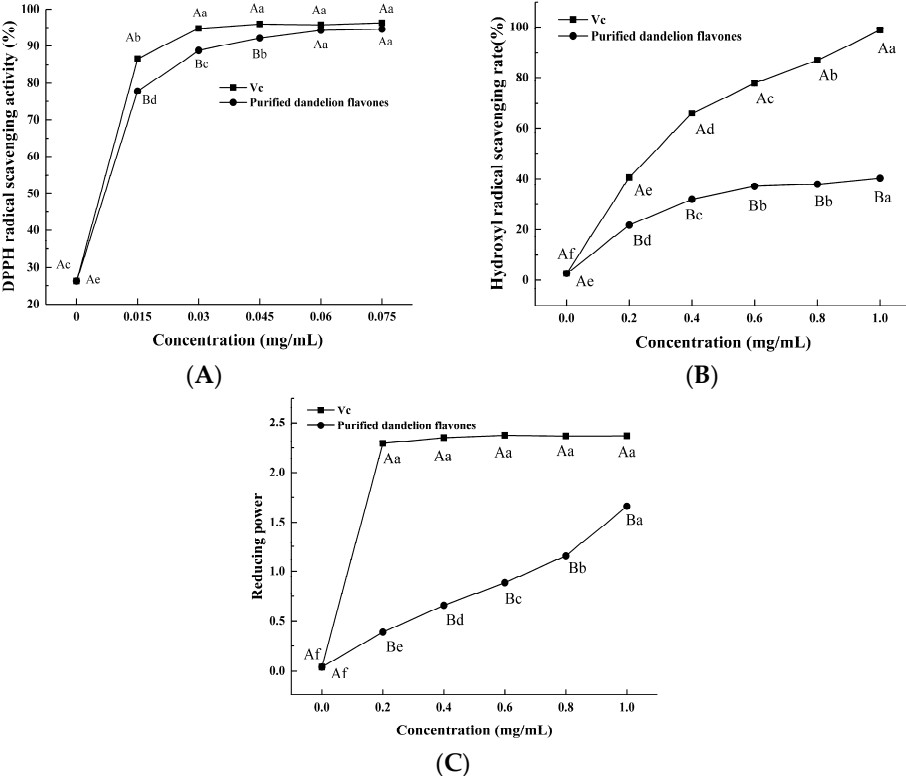

**Figure 6.** In vitro antioxidant activities of PFDF. (**A**) DPPH radical scavenging activity; (**B**) hydroxyl radical scavenging activity; (**C**) reducing power. Experiments were performed in triplicate, and data are expressed as the mean ± SE. Different uppercase superscript letters denote statistically significant differences between PFDF and VC ($p < 0.05$). Different lowercase superscript letters denote statistically significant differences among different concentrations ($p < 0.05$).

Hydroxyl radicals are the most active free radical, have the fastest reaction rate, and result in the greatest damage to biomolecules. Thus, the hydroxyl radical scavenging activity is frequently employed to assess a substance's potential to scavenge free radicals. The capacity of PFDF to scavenge hydroxyl radicals increased in a dose-dependent manner, but it was always lower than that of VC (Figure 6B). At a concentration of 1 mg/mL, the scavenging ability of PFDF was 40.26%. Thus, PFDF exhibited a capacity to donate H+ to interact with hydroxyl radicals and generate a stable form.

The reducing power of a substance is a measurement of its reductive ability as an antioxidant, and it is estimated by the transformation of ferric iron ($Fe^{3+}$) to ferrous iron ($Fe^{2+}$) in the presence of the sample extract [27]. The absorbance measured at a particular

wavelength can then reflect the strength of its reducing power; the more intense the absorption, the greater the reducing power [23,28]. The ability to reduce $Fe^{3+}$ can be attributed to the electron-donation capacity of phenolic compounds as described by Shimada et al. [29]. As a positive control, the VC for each concentration showed a higher reducing power than PFDF (Figure 6C). The reducing power of PFDF was amplified in our study with increasing concentration, exhibiting an effective capacity for donating electrons.

### 3.4. Intracellular ROS Production, Cell Death and Lipid Peroxidation of Zebrafish Embryo Treated with PFDF

Because of its tiny size, great spawning capacity, relative transparency, low cost, and physiologic similarities to mammals, the zebrafish has been employed as a vertebrate model organism [30]. Recently, the zebrafish model has been used as an alternative in vivo model to search and evaluate natural antioxidants.

ROS is a product of cellular metabolism and plays dual beneficial and deleterious roles in different organ systems [31]. The intracellular generation of ROS can be measured by analyzing the intensity levels of DCFH-DA fluorescence. As shown in Figure 6A,B, the control group without PFDF showed significantly augmented ROS production. However, zebrafish embryos pretreated with PFDF at doses of 1.25 and 2.5 µg/mL exhibited reduced ROS production ($p < 0.05$), respectively. These results suggest that PFDF might be a potential candidate to inhibit ROS formation and exert a protective action against oxidative stress through its antioxidant activity.

Damage to important cellular macromolecules such as lipids, proteins, and nucleic acids is induced by excess ROS [32]. Furthermore, excessive ROS production results in a variety of biochemical and physiological lesions such as cell injury and death. DPPP can pass through the cell membrane and cause oxidation via hydroperoxides produced by lipid peroxidation, resulting in luminous DPPP oxide. AO is a nucleic acid selective fluorescent cationic dye and is often used to identify apoptotic cells by interacting with DNA and RNA via intercalation or electrostatic attraction [19]. We dyed the zebrafish with DPPP and AO to assess the protective effects of PFDF against lipid peroxidation and cell death induced by ROS. As can be seen from Figure 7C,D, the addition of 1.25 and 2.5 µg/mL PFDF reduced the severity of lipid peroxidation ($p < 0.05$). Compared with the control group, all PFDF treatment groups showed drastically decreased cell death in a dose-dependent manner ($p < 0.05$) (Figure 7E,F). These results suggest that PFDF showed strong attenuated effects on lipid peroxidation and cell death in the zebrafish. The possible reason is that PFDF contains a large number of flavonoids as mentioned above.

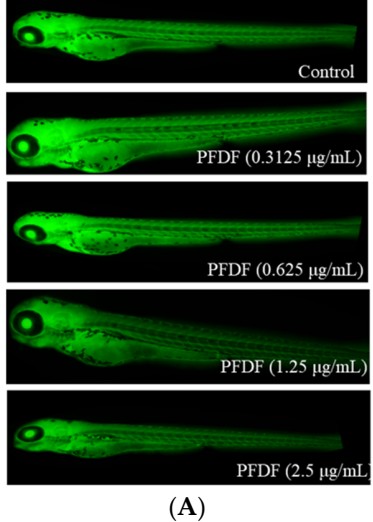

**(A)**

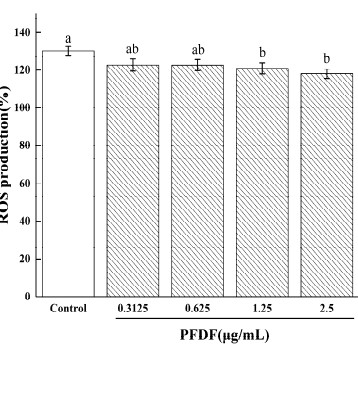

**(B)**

**Figure 7.** *Cont.*

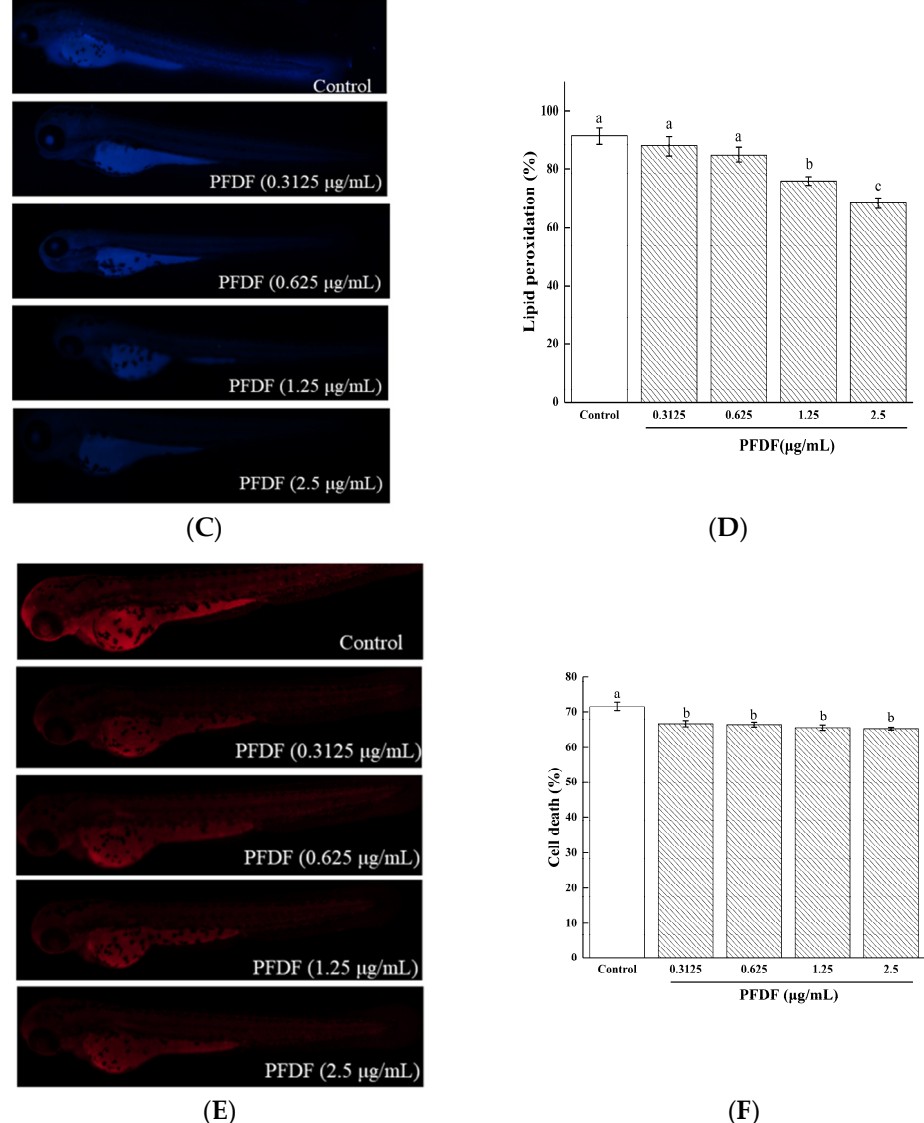

**Figure 7.** The effects of various concentrations of PFDF on ROS production (**A**,**B**), lipid peroxidation (**C**,**D**), and cell death (**E**,**F**) in zebrafish embryos measured by fluorescence microscopy and image J software. Experiments were performed in triplicate, and data are expressed as the mean ± SE. Different lowercase superscript letters denote statistically significant differences among different concentrations ($p < 0.05$).

### 3.5. Cell Proliferative Activity and MDA Content of IPEC-J2 Cells Treated with PFDF

Cellular proliferation is a process whereby cells grow and divide to replace those that have died or to expand a cell population. Our results showed that the proliferation activity of IPEC-J2 cells after the addition of 2.5 µg/mL PFDF was significantly increased compared to the control group, while proliferation activity at the remaining concentrations was not significantly altered (Table 3). In accordance with our study, it was found that dandelion extract protected rat skeletal muscle cells from the impact of LPS and promoted their proliferation [33]. Malondialdehyde (MDA) is a lipid peroxidation product caused by ROS that represents the degree of oxidative damage in living organisms [34]. In this study, the MDA content of the control group was significantly higher than that of the 2.5 and 5 µg/mL groups ($p < 0.05$), while the remaining groups showed no significant difference from the control group. In line with our study, Abdel-Magied et al. (2019) found that dandelion root extract can significantly reduce the MDA level in the liver and testes of

rats [35]. These results confirmed that PFDF could protect cells from oxidative damage by its strong antioxidative activity.

**Table 3.** Effect of PFDF on the proliferative activity and MDA content of IPEC-J2 cells.

| PFDF (µg/mL) | Proliferative Activity of IPEC-J2 Cells (%) | Malondialdehyde (nmol/mg Prot) |
|---|---|---|
| control | 100 [bc] | 0.53 [a] |
| 0.625 | 97.31 [c] | 0.41 [ab] |
| 1.25 | 102.10 [abc] | 0.37 [ab] |
| 2.5 | 122.11 [a] | 0.23 [b] |
| 5 | 120.49 [ab] | 0.28 [b] |
| 10 | 96.10 [c] | 0.42 [ab] |
| SEM | 2.40 | 0.02 |
| *p*-value | 0.0296 | 0.0248 |

Experiments were performed in triplicate, and data are expressed as the mean $\pm$ SE. Different lowercase superscript letters denote statistically significant differences among different concentrations ($p < 0.05$).

*3.6. Antioxidant Enzyme Activities and GSH Content of Zebrafish Embryo and IPEC-J2 Cells Treated with PFDF*

The body's ROS molecules are eliminated in a state of dynamic equilibrium under normal conditions; but, when the body is under oxidative stress, the oxidative and antioxidative changes produce an imbalance. As a result, intracellular ROS levels rise to cytotoxic levels, potentially resulting in oxidative DNA damage, aberrant protein expression, and tissue damage. Living organisms can relieve oxidative stress via enzymatic and non-enzymatic defense systems. As the enzymatic defense systems, SOD catalyzes the dismutase reaction of superoxide anion radicals into $H_2O_2$, and then CAT converts $H_2O_2$ into $H_2O$ and oxygen. In addition, GSH-Px works on GSH to reduce $H_2O_2$ and lipid hydroperoxides to $H_2O$ and the corresponding alcohols [36,37]. In this study, we evaluated the antioxidative enzymes' (SOD, CAT, and GSH-Px) activity in zebrafish embryos. As shown in Table 4, GSH-Px activity was increased in zebrafish embryos pretreated with 1.25 or 2.5 g/mL PFDF ($p < 0.05$). However, there was no significant difference in SOD or CAT activity between the PFDF treatment groups and the control group. According to this finding, it could be suggested that PFDF substantially reduced cell death, ROS production, and lipid peroxidation in zebrafish by enhancing the antioxidative enzyme activities.

**Table 4.** Effect of PFDF on the antioxidant enzyme activities of zebrafish embryos.

| PFDF (µg/mL) | Superoxide Dismutase (U/mg Prot) | Catalase (U/mg Prot) | Glutathione Peroxidase (U/mg Prot) |
|---|---|---|---|
| control | 16.88 | 7.13 | 1.43 [a] |
| 0.3125 | 16.88 | 4.27 | 5.98 [a] |
| 0.625 | 15.05 | 6.11 | 4.29 [a] |
| 1.25 | 16.64 | 7.61 | 21.23 [b] |
| 2.5 | 15.77 | 7.41 | 18.81 [b] |
| SEM | 0.36 | 0.62 | 4.03 |
| *p*-value | 0.4186 | 0.1808 | 0.0011 |

Experiments were performed in triplicate, and data are expressed as the mean $\pm$ SE. Different lowercase superscript letters denote statistically significant differences among different concentrations ($p < 0.05$).

The antioxidative capacities of dandelion extracts were proven in RAW264.7 cells and the human platelets model with enhanced antioxidative enzyme activities and the relief of oxidative damage under diverse stimuli-induced oxidative stressors [7,38,39]. In line with previous observations, the intracellular SOD and CAT activities of IPEC-J2 cells without PFDF were significantly lower than those of cells treated with 5 and 10 µg/mL PFDF ($p < 0.05$) (Table 5). However, no significant difference in GSH-Px activity was found among different treatments. The GSH-Px enzyme in IPEC-J2 cells could be less

susceptible to PFDF. It is suggested that the activity of GSH-Px is highly dependent on the quantity of selenium in the blood or tissue since it is a selenoenzyme [40]. Furthermore, a previous study reported that the antioxidants, such as polysaccharides from dandelion roots, could enhance SOD and CAT activities as well as promote the GSH level to protect tissue from oxidative damage [41]. GSH has several biochemical functions, including removing oxygen ions and other free radicals, boosting enzymes' activities, and acting as an antioxidant. GSH is the most important member of the non-enzymatic defense systems in living organisms [42]. To further investigate the protective effects of PFDF, the GSH content of IPEC-J2 cells was determined. It was found that the GSH content was significantly higher in all treatment groups except for the group with 10 µg/mL PFDF ($p < 0.05$). Therefore, PFDF could have promoted proliferation and reduced the MDA content of IPEC-J2 cells by increasing the antioxidative enzyme activities and GSH content.

**Table 5.** Effect of PFDF on antioxidant enzyme activities and GSH level in IPEC-J2 cells.

| PFDF (µg/mL) | Superoxide Dismutase (U/mg Prot) | Catalase (U/mg Prot) | Glutathione Peroxidase (U/mg Prot) | Glutathione (µmol/g Prot) |
|---|---|---|---|---|
| control | 3.18 [b] | 21.42 [b] | 226.27 | 3.63 [c] |
| 0.625 | 4.57 [b] | 22.64 [b] | 205.80 | 4.84 [ab] |
| 1.25 | 5.68 [ab] | 25.43 [b] | 250.25 | 4.97 [ab] |
| 2.5 | 6.15 [ab] | 25.97 [b] | 235.13 | 5.10 [ab] |
| 5 | 8.60 [a] | 34.14 [a] | 387.27 | 5.46 [a] |
| 10 | 8.32 [a] | 33.57 [a] | 371.65 | 4.11 [bc] |
| SEM | 0.38 | 0.77 | 12.96 | 0.11 |
| *p*-value | 0.0073 | 0.0014 | 0.1234 | 0.0288 |

Experiments were performed in triplicate, and data are expressed as the mean ± SE. Different lowercase superscript letters denote statistically significant differences among different concentrations ($p < 0.05$).

## 4. Conclusions

In summary, the optimized parameters for the purification process of flavonoids on AB-8 macroporous resin were a sample concentration of 2 mg/mL, pH of 2.0, and sample volume of 120 mL; an elution solvent ethanol–water (70%), pH of 5, and eluent volume of 90 mL. PFDF was found to be enriched in flavonoid glycosides and flavonoid aglycones and exhibited antioxidative effects on IPCE-J2 cells and zebrafish embryos, evidenced by reduced oxidative damage and adjusted antioxidant enzymes activities. Therefore, we suggest that PFDF can be used as a potential natural and functional food additive to prevent oxidative stress-induced damage. However, the underlying mechanism of PFDF regulating antioxidant enzymes' activities is still unclear and should be analyzed.

**Supplementary Materials:** The following supporting information can be downloaded at: https://www.mdpi.com/article/10.3390/fermentation8070306/s1, Table S1. The relative content of major flavonoid glycosides; Table S2. The relative content of major flavonoid aglycones.

**Author Contributions:** Conceptualization, X.A. and J.Q.; methodology, N.Y. and Y.W.; software, N.L. and Y.Z.; validation, Y.W.; investigation, N.Y., X.R., Y.W., Y.Z. and N.L.; resources, X.A. and J.Q.; data curation, N.Y., Y.W. and X.R.; writing—original draft preparation, N.Y. and Y.W.; writing—review and editing, Y.W.; supervision, X.A. and J.Q.; project administration, J.Q.; funding acquisition, J.Q., X.A. and Y.W. All authors have read and agreed to the published version of the manuscript.

**Funding:** This research was funded by the Major Science and Technology Program of Inner Mongolia Autonomous Region, grant number 2021ZD0024-4, Major Science and Technology Program of Inner Mongolia Autonomous Region, grant number 2021ZD0023-3, Major Science and Technology Program of Inner Mongolia Autonomous Region, grant number 2020ZD0004, Key Technology Project of Inner Mongolia Autonomous Region, grant number 2020GG0030, and Scientific and Technological Achievements Transformation Project of IMAU, grant number YZGC2017025.

**Institutional Review Board Statement:** Not applicable.

**Informed Consent Statement:** Not applicable.

**Data Availability Statement:** Not applicable.

**Conflicts of Interest:** The authors declare no conflict of interest.

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
