# Peer review of "Isolation and Characterization of Flavonoids from Fermented Dandelion (Taraxacum mongolicum Hand.-Mazz.), and Assessment of Its Antioxidant Actions In Vitro and In Vivo"

_fermentation, doi:10.3390/fermentation8070306_

Round 1
Reviewer 1 Report
The authors carried out the fermentation of dandelion with a yeast-LAB culture, and then the isolation and characterization of flavonoids were performed. The research idea is original, but at the moment it can not be accepted in the present form. Comments and suggestions on fermentation-1746888:
- I believe that the characterization of flavonoids in unfermented dandelions should be also provided in the current study to compare the beneficial effects of fermentation.
- Please provide more information about bioactive compounds in dandelions (introduction part).
- The aim of the work should be emphasized and the statement of the novelty should appear.
- The methodology of preparation of PFDF - On what basis were the fermentation conditions selected? Also, why were such amounts of water or alcohol as well as time and temperature set?
- Subsection 2.7.3. should be more specified.
- The information about supplementary materials should be provided in the manuscript. According to the instructions for authors the following information should be stated: “Supplementary Materials: The following supporting information can be downloaded at: www.mdpi.com/xxx/s1, Figure S1: title; Table S1: title; Video S1: title.” Moreover, in the file with the supplementary materials the tables should be named, i.e. Table S1, Table S2, etc. Please indicate also in the manuscript description of the tables.
- The values given in the text do not always correspond to the values in figures 6B, 6D, and 6F.
- All the abbreviations that appeared in the tables should be defined in the table footers.
- The manuscript has a very weak discussion of the results and it should be definitely improved.
Minor suggestions:
- Line 58 - The more common name of the family to that dandelions belong is the family Asteraceae.
- Line 58 – delete “(dandelion)”
- Lines 67, 99-100, 114-115, etc. - use italics for microorganisms' name
- Line 100 – the proper name for the L. plantarum should be used according to the new nomenclature.
- Line 103-106, 153, etc. – use the subscripts in the structures of used chemicals.
- Line 116 – improve the number of microorganisms.
- Line 213 – Artemia salina should be written in italics.
- Line 321 – improve the sentence
- “O” in the names of the chemicals should be written in italics.
- Line 354 – “VC” should be defined
- The reference list should be prepared according to the instructions for the authors.
Reviewer 2 Report
Dear Editor, dear Authors,
I have carefully read the manuscript entitled “Isolation and characterization of flavonoids from fermented dandelion (Taraxacum mongolicum Hand.-Mazz.), and assessment of its antioxidant actions in vitro and in vivo” and find it very insightful and significant for the scientific community. The manuscript is interesting, well written and well structured, the available scientific data are summarized and discussed clearly.
I suggest that the paper should be accepted after some minor corrections listed bellow:
*** In the whole text, in vivo and in vitro must be italic ***
Introduction
- line 67-68 names of microorganisms must be italic!
- expand the part about potential of fermentation of dandelions in this part
- avoid using “we” and “our” through the whole paper
Materials and Methods
- line 99-100, line 114 – names of microorganisms must be italic!
- Authors should define the procedure for preparation of microorganisms for inoculum, determination of concentration, and all further steps which were involved in the experiment
- line 263 – reference is not presented as reference number
Results
- in all analyses were done in duplicate; the obtained results must be presented as mean ± st. dev.
Conclusions
This part needs to be expanded with all necessary details, and further steps
Round 2
Reviewer 1 Report
The authors addressed all the comments and suggestions, as well as, carefully improved the manuscript. I have only one more suggestion - in Tables 1 & 2 please use the subscripts in the chemical formulas.